# Effectiveness of Nature-Based Solutions for Mitigating the Impact of Pluvial Flooding in Urban Areas at the Regional Scale

**Eike M. Hamers [1,*], Holger R. Maier [1], Aaron C. Zecchin [1] and Hedwig van Delden [1,2]**

1   School of Architecture and Civil Engineering, The University of Adelaide, Adelaide, SA 5005, Australia
2   Research Institute for Knowledge Systems (RIKS), P.O. Box 463, 6200 AL Maastricht, The Netherlands
*   Correspondence: eike.hamers@tuhh.de

**Abstract:** Pluvial flooding causes significant damage in urban areas worldwide. The most common approaches to mitigating these impacts at regional scales include structural measures such as dams, levees and floodways. More recently, the use of nature-based solutions (NBS) is receiving increasing attention, as such approaches are more adaptive than structural measures and have a number of potential co-benefits (e.g., improvements in water quality and amenity). As NBSs are generally applied at house or block scales in urban areas, their potential for reducing the impacts of urban flooding at the regional scale are unknown. We introduce an approach that enables the potential of using portfolios of NBSs to reduce the impact of urban flooding to be assessed at the regional scale. This approach enables the most suitable locations for such portfolios of NBSs to be identified, as well as their effectiveness to be modeled at spatial resolutions that are commonly used for regional planning studies. The approach is applied to a case study area to the north of Adelaide, South Australia, with results obtained suggesting that there is significant potential for using strategically placed portfolios of NBSs to reduce the impact of pluvial flooding in urban areas at the regional scale.

**Keywords:** nature-based solutions; hydrodynamic modeling; urban flood management; urban flooding; flood risk; land use planning

## 1. Introduction

With more than 1500 documented catastrophic flood events occurring worldwide from 2010 to 2020, resulting in overall damages costing USD 363 Billion, flooding is one of the most expensive natural hazards [1,2]. The causes of floods can be quite different, but heavy rainfall is often a key factor in the emergence of a flood. Extreme rainfall events on a catchment cause stress on the river network downstream and a significant rise in overland flow within the affected region [3–5].

With an increase in urbanization around the world, the risk of severe flooding caused by heavy rainfall is increasing significantly as a result of an increase in impervious areas [6,7]. These newly built areas also have a significantly higher building stock and asset value than rural regions. Consequently, there has been a significant increase in flood risk due to increases in exposure [5,8]. This trend is likely to increase in the future. For example, if urbanization rates continue without change, in 2050, the expected global increase in sealed surfaces in urban areas of up to 15,000 km$^2$ will lead to even larger runoff and, therefore, even greater flood extents and greater water depths. At the same time, globally, USD 53 trillion of value is expected to be added to the 1 in 100-year floodplain by 2050, which would lead to an accumulated value of USD 80 trillion [3,9]. In addition, rainfall events are also likely to become more intense due to the impacts of climate change [10,11], further increasing flood risk.

With the expected increases in flood risk outlined above, demands for flood mitigation are also likely to increase into the future. Structural mitigation strategies, such as dams and levees, belong to the class of gray infrastructure and are arguably the most widely used mitigation options at present. For example, White [12] used the implementation of structural

mitigation to investigate human adjustment to flood risk in the United States of America, whereas Birkland et al. [13] and Thampapillai and Musgrave [14] considered the impact of structural mitigation on the environment. As demonstrated by the Australian Institute for Disaster Resilience [15] in the case of riverine flooding, structural interventions are able to direct the flow path of floods away from areas of higher value towards areas of lower value and can be effective over large areas, enabling flood risk to be reduced at regional scales. However, despite their proven effectiveness, structural measures also have disadvantages. For example, as shown by Birkland et al. [13] for the case of structural mitigation and Thampapillai and Musgrave [14] when considering traditional mitigation, these measures are generally expensive to construct and maintain. In addition, as demonstrated by Thampapillai and Musgrave [14] through a review of different flood mitigation measures, structural measures are generally not well suited to adaptation once constructed, making them less able to respond to unknown changes in future conditions. Finally, structural mitigation can also have a negative impact on the environment, as demonstrated by Birkland et al. [13], who showed that such measures influence both upstream and downstream ecosystems.

In response, there has been increasing interest in the development of more adaptive flood mitigation options [16–25]. In contrast with structural measures, many of the options that are considered to be more adaptive belong to the class of green and blue infrastructure, or nature-based solutions (NBSs), which use natural elements to reduce water depths and flow speeds within a certain area by mimicking the effects of vegetation and soil characteristics on floods and their distribution [26–30]. This is achieved by increasing surface roughness and permeability, thereby reducing the impact of localized flooding by reducing flow velocities and increasing infiltration rates. Examples of NBSs include rain gardens, green roofs, retention basins, wetlands, and re-naturalized river systems; these are often used in conjunction with some sort of storage facility to further reduce flood peaks [24,31–34]. In addition to mitigating flood risk, NBSs are also able to increase amenities in urban areas [23,30].

In contrast with structural mitigation options, the effectiveness of NBSs in urban areas has generally only been assessed at smaller, localized scales, such as street blocks or small suburbs [27,28,35–38], with assessments at larger scales being rare [33,39,40]. Given these localized assessment scales, previous studies have generally focused on the detailed modeling of the effectiveness of individual NBS schemes at known locations, investigating the relative effectiveness of different types of NBSs under different rainfall regimes, including the impact of climate change [27]. However, the effectiveness of applying portfolios of NBSs at regional scales (e.g., combinations of different NBSs such as green roofs, swales, rain gardens, retention basins, wetlands, etc., that are distributed over an extended area and considered as a single, combined solution) to complement (or act as potential alternatives to) structural measures, has not been considered, despite the potential benefit they could provide in terms of increased adaptability and amenity, as well as reduced costs.

A likely reason for the lack of consideration of the potential benefits of using portfolios of NBSs for urban flood mitigation at regional scales is that there is currently no formal approach to determining how many NBSs to use and where to locate them to achieve an appropriate trade-off between the number of NBSs (and hence their costs) and the corresponding reduction in flood impact. In addition, assessment of the relative effectiveness of different portfolios of NBSs requires the development of a modeling approach that is suited to regional scale assessments, which is also not available at present. When considering regional scales, a coarser modeling resolution is more appropriate, as the focus is on the identification of the most suitable locations of NBSs, rather than the detailed modeling of individual schemes at a given location. Such a coarser resolution is likely to facilitate better integration with the land-use maps and models required to determine the suitability of different potential locations of NBSs and to enable different placement configurations to be modeled in a computationally efficient manner.

In order to address the above limitations, the objectives of this paper are:

(1)  To develop a formal approach that is able to: (i) identify suitable locations of portfolios of NBSs at regional scales, enabling trade-offs between portfolio size and the corresponding reduction in urban flood impact to be determined; and (ii) model the flood reduction impact of portfolios of NBSs at regional scales at a resolution that enables the requisite analyses to be integrated with land use planning practices and to be conducted in a computationally efficient manner.

(2)  To illustrate the utility of the proposed approach by assessing the degree to which portfolios of nature-based solutions can mitigate urban flooding at the catchment scale for a case study in Adelaide, South Australia.

The remainder of this paper is organized as follows. The proposed methodology for identifying and modeling the most suitable locations of portfolios of NBSs at the regional scale is introduced in Section 2, followed by the case study application in Section 3. The results are presented and discussed in Section 4 and the conclusions are provided in Section 5.

## 2. Methodology

In this section, the proposed approach for including portfolios of NBSs as potential options in regional flood reduction studies is introduced. This includes (i) identification of where portfolios of NBSs should be placed throughout a region (Objective 1 (i)—see Section 2.1) and (ii) how to best model the impact of NBSs at regional, rather than street or small suburban scales (Objective 1 (ii)—see Section 2.2).

### 2.1. Placement of Portfolios of NBSs at Regional Scales

The proposed approach to identifying relevant locations of portfolios of NBSs at regional scales to enable trade-offs between portfolio size and the corresponding reduction in flood impact to be determined is outlined in Figure 1. As can be seen, the proposed approach consists of four main steps, including determination of potential locations for NBSs (Step 1, Figure 1), determination of relevant locations for NBSs (Step 2, Figure 1), determination of the placement of NBSs (Step 3, Figure 1), and determination of the effectiveness of the NBSs placed as part of Step 3 (Step 4, Figure 1). Knowledge of the relative effectiveness of the placement of NBSs in different catchment regions can potentially be used to refine the placement of NBSs to achieve better trade-offs between the number of NBSs and the corresponding reduction in flood risk as part of an iterative formal or informal optimization process [41].

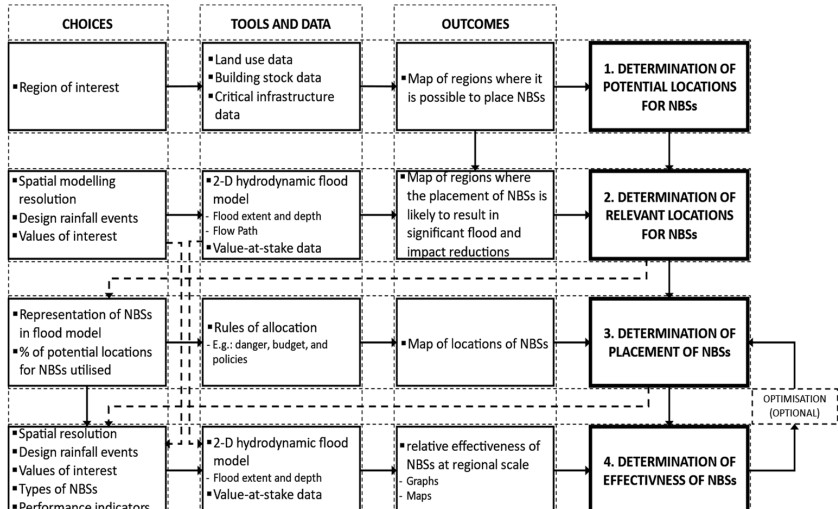

**Figure 1.** Generic approach for placing nature-based solutions throughout a catchment in a regional urban flood mitigation planning setting.

For each of the four steps mentioned above, an outline of the choices to be made by users of the approach, the tools and data required, and the outcomes obtained are also given in Figure 1. Steps 2, 3, and 4 involve the modeling of pluvial flooding in the area of interest to determine the extent of the flood and the impact of NBSs. Details of each of the four steps are given in subsequent paragraphs.

In order to identify potential locations for the placement of portfolios of NBSs (Step 1, Figure 1), the region for which there is interest in determining the potential of using portfolios of NBSs to reduce flood risk has to be defined, along with areas in which it is possible to locate NBSs. The latter requires information on the spatial distribution of different land use classes, as well as the location of building stock and critical infrastructure. Other local, context-specific information (e.g., local regulations, social factors, etc.) should also be considered. The outcome of this step is a map of regions where it is possible to place NBSs.

As part of the next step (Step 2, Figure 1), regions in which the placement of portfolios of NBSs is likely to result in a significant reduction in flood levels and impacts are identified. This is achieved by examining the coincidence of the potential locations of portfolios of NBSs identified in Step 1 with flood extent and depth and value-at-stake data (i.e., values of buildings, critical infrastructure, ecological assets, etc.). Choices to be made in this step include the design rainfall events against which the effectiveness of nature-based solutions is going to be tested, as well as the values of interest to be included (e.g., economic, social, environmental, etc.). The flood maps that correspond to the selected rainfall events can either be obtained from previous studies or with the aid of an appropriate model (e.g., 1-D/2-D hydrodynamic flood model) and should contain information such as flood depth, flood extent and flow velocity. The outcome of this step is a map of the relative feasibility of potential locations of NBSs in terms of flood risk reduction.

As part of the third step (Step 3, Figure 1), locations where portfolios of NBSs should be placed are determined. This is achieved with the aid of rules of allocation that consider a combination of factors, such as the map of the relative feasibility of potential locations of NBSs obtained as part of Step 2, the types of NBSs to be considered, the available budget/number of NBSs/fraction of relevant locations to be utilized, local policies/plans/restrictions, etc. These rules are case-study specific and need to be predetermined. For example, inundation depth and land use cover can be used to determine the different allocations. As part of high-level planning exercises, such an approach could be automated based on numerical criteria (e.g., inundation level thresholds), although in practice would most likely also include stakeholder engagement processes (see Di Matteo, Dandy [42], Di Matteo, Maier [43], Wu, Maier [44]). The outcome of this step is a map of the locations of portfolios of NBSs to be considered.

The fourth step (Step 4, Figure 1) involves determination of the effectiveness of the configuration of NBSs selected in Step 3 in terms of the performance indicators of interest, such as reduction in flood depth and extent and/or reduction in flood damage. This is achieved with the aid of a flood model that enables the impact of the selected configuration of NBSs on the selected performance indicators to be assessed (e.g., 1-D/2-D hydrodynamic model). If these indicators (e.g., damage to building stock) go beyond purely hydrodynamic factors, additional information translating hydraulic variables to the required impact metrics is also required (e.g., which buildings are inundated, the values of these buildings and the vulnerability curves translating inundation levels to the degree of building damage).

When developing the hydraulic model used to assess the effectiveness of the selected configuration of NBSs, it is critical that an appropriate spatial modeling resolution is used. The resolution that is most appropriate depends on a number of factors, such as the modeling approach used, the available computational resources, the scale of the NBSs considered, and the spatial resolution at which planning studies are conducted. Given that the spatial modeling resolution is likely to be significantly larger than the spatial extent of individual NBSs (see Section 1), a key issue that needs to be addressed is how to best represent portfolios of NBSs in the flood simulation model so that the impact of the

addition of portfolios of NBSs at the regional scale can be assessed with an appropriate level of accuracy and computational efficiency (Objective 1(ii)). The proposed approach for achieving this is outlined in Section 2.2.

In practice, it is unlikely that only a single configuration of portfolios of NBSs will be assessed. For example, there might be interest in repeating Steps 3 and 4 for different numbers of portfolios of NBSs/percentage utilization of relevant locations (and hence cost), enabling trade-offs between the number of NBSs and corresponding performance indicators to be determined. Alternatively, formal optimization algorithms could be used to determine the configurations of portfolios of NBSs that result in the optimal trade-offs between the number of NBSs considered (and hence cost) and flood risk reduction (as measured by the selected performance indicators) (e.g., see Di Matteo, Maier [43]). Consequently, potential outcomes of this step include graphs of (optimal) trade-offs between the number of NBSs and flood risk reduction and/or maps of the relative flood risk reduction resulting from different configurations of NBSs.

### 2.2. Modeling of Impact of NBSs at Regional Scales

As mentioned in Section 1, in previous studies, the effectiveness of NBSs has been assessed by modeling the impact of a particular type of NBS at a particular location, rather than modeling the impact of a portfolio of NBSs at regional scales. In order to achieve the latter, the modeling resolution (i) has to be commensurate with that used in regional land use planning studies (e.g., Newland, van Delden [45], Newland, Maier [46]) and (ii) result in a model that is sufficiently computationally efficient to enable the relative effectiveness of different portfolios of NBSs to be assessed in a reasonable timeframe.

In order to achieve this, it is proposed to: (i) adopt a spatial resolution that is appropriate for the case study under consideration, considering factors such as the spatial extent of the area to be modeled, the modeling approach used, the available computational resources, the scale of the NBSs considered, and the spatial resolution at which planning studies are commonly conducted (e.g., from 50 m $\times$ 50 m to 500 m $\times$ 500 m) (see Section 2.1); and (ii) use an equivalent uniform infiltration rate for each of these spatial areas, which is a function of the number, type, and extent of NBSs on this area, rather than modeling each scheme individually in a prohibitively detailed manner. The uniform infiltration rate simplifies the otherwise complex modeling of NBSs to allow for an automated allocation in larger regions.

To determine an appropriate uniform infiltration rate for the selected spatial modeling resolution and the types of NBSs considered, the "calibration" approach depicted in Figure 2 is proposed. As part of the approach, typical numbers of building- (e.g., green roofs, rain gardens—Scheme A, Figure 2) and block-size (e.g., wetlands—Scheme B, Figure 2) measures are selected for a single spatial modeling unit at representative locations in the region of interest. Runoff hydrographs resulting from different spatial configurations of these portfolios of NBSs are then obtained at numerous locations within this spatial unit with the aid of the flood simulation model selected in Step 2. These runoff hydrographs are compared with the hydrographs obtained by applying an "equivalent" uniform infiltration rate over the same spatial modeling unit using the same flood simulation model, which is adjusted in an iterative fashion until a satisfactory match is obtained between the hydrographs obtained by modeling individual portfolios of NBSs and those obtained by applying an equivalent uniform infiltration rate (Figure 2).

How well these two sets of hydrographs match can be assessed using visual inspection or in a more quantitative fashion using a range of performance metrics (see Bennett, Croke [47]). The process of iteratively adjusting the equivalent infiltration rate can be carried out manually or using more formal optimization methods (see Maier, Razavi [41]). The convergence of this iterative process is likely to increase by starting the iterative process with a value that is informed by an understanding of the underlying processes (e.g., see Newland, van Delden [45]; Bi, Maier [48]), such as using the area-weighted average of the infiltration values of the individual members of the portfolio of NBSs considered.

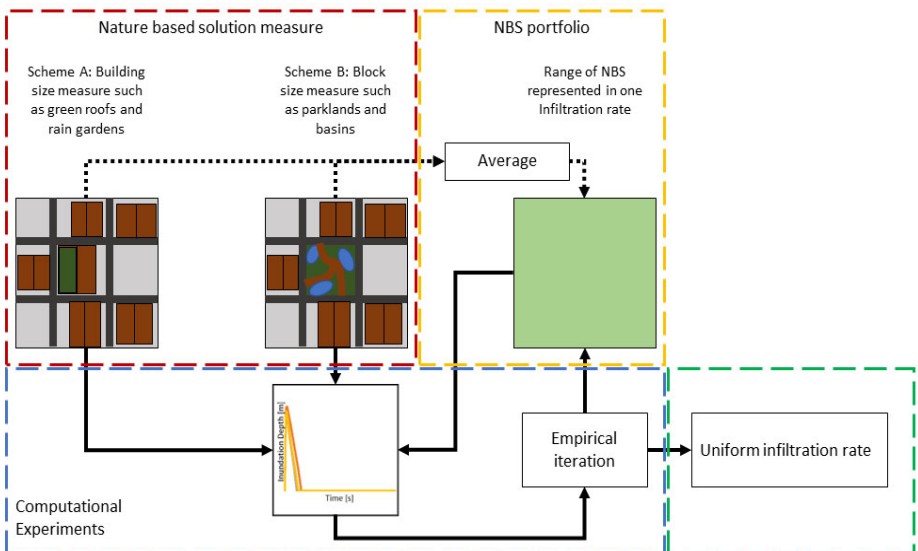

**Figure 2.** Methodology to determine uniform infiltration rate for NBS portfolio.

## 3. Case Study

In this section, the methodology used to (i) illustrate the utility of the approach introduced in Section 2 and (ii) assess the degree to which different portfolios of nature-based solutions can mitigate urban pluvial flooding at the catchment scale is outlined for a case study in Adelaide, South Australia (Objective 2). The choices made, tools/data used, and outcomes achieved for each of the four steps of the proposed methodology for the case study are detailed in the following sub-sections.

### 3.1. Determination of Potential Locations for NBSs

The Gawler River region to the north of Adelaide, South Australia, was selected as the case study area (Figure 3). The region covers an area of 683.22 km$^2$ and spans seven different local government areas. The majority of the region is considered to be rural, mostly consisting of agri- and horticultural- development and low-density rural residential areas [49]. However, there are also urban areas with high density development, especially around the township of Gawler.

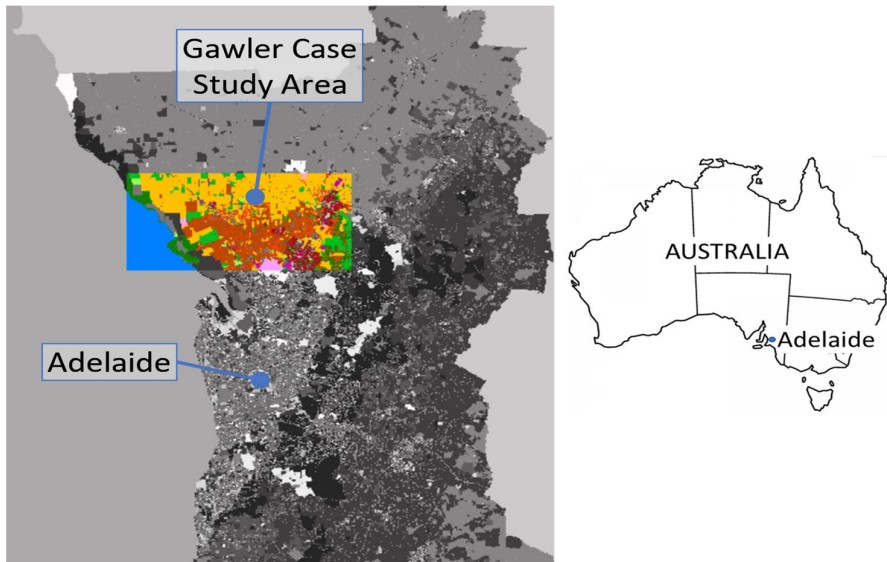

**Figure 3.** Gawler case study area in the North of the Greater Adelaide region.

The region is affected by significant flooding. The most recent major event in 2016 caused AUD 50 million in damage and resulted in the formation of the Gawler River Floodplain Management Authority, as well as planning for a floodway along the river to protect the most vulnerable areas [49,50]. However, there has been no consideration of using NBSs at the regional scale to reduce flood risk.

Land use and building stock maps were available at a 100 m × 100 m resolution and were used to identify potential locations for NBSs, which corresponded to residential, commercial, and industrial land use classes (see Figure 4, Step 1, Tools and Data). This resulted in a map of regions in which it would be possible to place NBSs (Figure 4, Step 1, Outcomes, red areas). As can be seen, these regions are primarily distributed in the eastern part of the catchment, with smaller pockets of potential locations for NBSs distributed throughout the remainder of the area. Overall, the potential areas for the placement of NBSs cover an area of 38.69 km², which corresponds to 6.66% of the total area considered. Details of the land use map and a map of potential locations are provided in Appendix A and Appendix C.

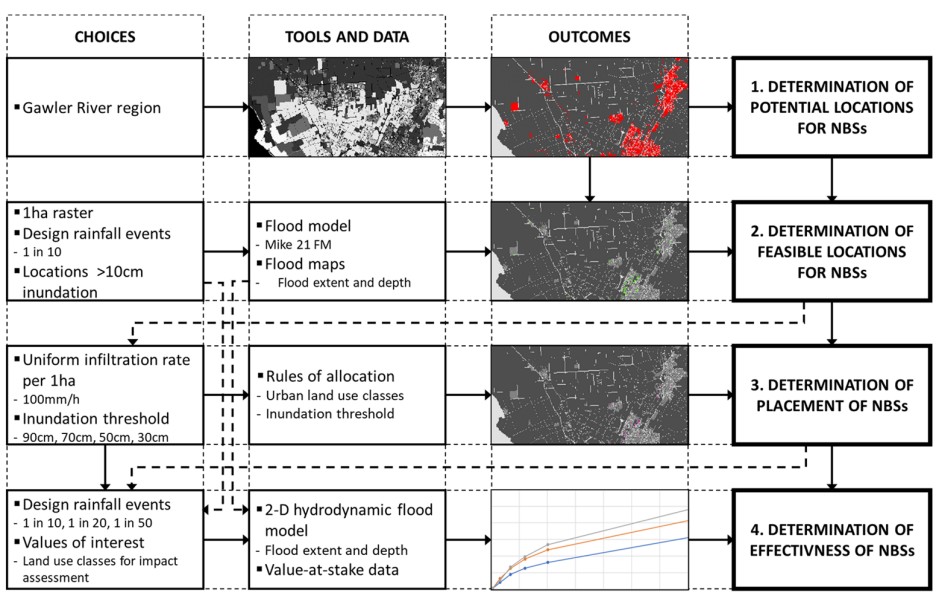

**Figure 4.** Application of proposed approach to the placement of nature-based solutions throughout the Gawler River case study region.

### 3.2. Determination of Relevant Locations for NBSs

In order to identify the relevant locations for NBSs, the map of potential locations for NBSs obtained in Section 3.1 was overlayed with the flood map for the 1:10 year pluvial flood event. Any potential locations that exceeded an inundation threshold (i.e., depth in excess of a selected value) of 100 mm were selected as relevant locations for NBSs. A return period of 10 years was selected, as this is a common return period when using NBSs [51], and an inundation threshold of 100 mm was used because this is the starting water depth for non-zero values on the depth–damage curves. This resulted in a map of relevant locations for placing NBSs (Figure 4, Step 2, Outcomes, green areas). These areas form a small subset of the potential areas, covering an area of 1.17 km², which corresponds to 3.02% of the Potential Area identified in Section 3.1 and 0.20% of the total area considered. Details of the map of potential locations, the 1:10 year inundation map and the map of relevant locations are given in Appendix B and Appendix C.

The inundation map was obtained with the aid of a 2D hydrodynamic flood model of the catchment developed using the software Mike Flood 21 FM [52]. The model was implemented using a flexible mesh and Digital Elevation Model (DEM) to simulate flood extent and flow throughout the catchment. The flexible mesh reduces computational

time significantly compared with using a standard grid, as this enables use of a variable resolution in computational points across the landscape and is designed for integration with graphics processing units (GPUs) (e.g., for the case study considered, this reduced run time from approximately 5 h to less than 1 h). A maximum area of 225 m$^2$ and an angle of 25° were used to create the mesh. However, it should be noted that input data on infiltration, roughness, and rainfall can be entered in a regular grid format, which was selected to be 15 m × 15 m to be commensurate with the grid sizes used for the detailed assessment of individual NBSs (e.g., Zölch, Henze [27], Huang, Tian [34], Vojinovic, Alves [35], Meshram, Ilderomi [53]). Both infiltration and rainfall rates are expressed in mm/h, and while the infiltration rate is spatially explicit, the rainfall rate is uniform throughout the area of interest.

Infiltration rates were set to 1 mm/h and 2 mm/h for urban and rural land use classes, respectively. This is because a reduction in infiltration across the catchment is assumed to be caused by extensive rainfall prior to the flood event in accordance with Tonkin Consulting [49]. In addition, the infiltration rates for cells corresponding to high degrees of surface sealing, such as those occupied by infrastructure, were set to 0 mm/h.

The model was calibrated and validated by comparing results with those of a published flood study conducted by Tonkin Consulting [49] for a 1 in 100-year event. The model is capable of simulating fluvial and pluvial flooding. Further details of the calibration and validation process are given in Appendix D.

### 3.3. Determination of Placement of NBSs

The allocation rules used to determine in which sub-areas of the relevant regions (identified in Section 3.2) to place portfolios of NBSs were based on different inundation thresholds, including 300 mm, 500 mm, 700 mm, and 900 mm. This was done for illustrative purposes, as it enabled the identification of different regions and extents where NBSs are placed (see Table 1), enabling trade-offs between different numbers of NBSs (and hence costs) and allowing the corresponding flood risk to be investigated. The impact of using an additional sub-region consisting only of urban residential land use classes (restricted potential, Table 1) was also investigated to narrow the gap between potential and relevant location coverages. As mentioned in Section 2.1, in practice, the allocation process would most likely also involve consideration of a range of other factors, such as local regulations and input from relevant stakeholders.

**Table 1.** Placement choices, area covered in NBS, and utilization based on different thresholds.

| Chosen Thresholds [cm] | 90 | 70 | 50 | 30 | Relevant | Restricted Potential | Potential |
|---|---|---|---|---|---|---|---|
| Area [km$^2$] | 0.03 | 0.18 | 0.40 | 0.70 | 1.17 | 19.38 | 38.69 |
| Utilization of catchment | 0.005% | 0.03% | 0.07% | 0.12% | 0.20% | 3.34% | 6.66% |
| Utilization of potential | 0.08% | 0.48% | 1.03% | 1.81% | 3.02% | 50.1% | 100% |
| Utilization of relevant | 2.48% | 15.8% | 34.1% | 59.9% | 100% | - | - |

As can be seen from Table 1, the restricted potential area covers just over half the area of the potential locations, whereas the relevant area only covers 3.02% of the potential area. As the required inundation threshold increases from 30 cm to 90 cm, the percentage of the potential area in which portfolios of NBSs are placed in accordance with the rules of allocation used reduces from 1.81% to 0.08%, corresponding to 59.9% and 2.48% coverage of the relevant region, respectively. This enables the importance of placing NBSs at strategic locations at the regional scale, and hence the potential utility of the proposed approach, to be assessed.

The outcome of the allocation process is the development of maps showing the location of NBSs for the different inundation thresholds considered (Figure 4, Step 3, Outcomes, pink areas), details of which are given in Appendix C.

### 3.4. Determination of Effectiveness of NBSs

In order to determine the effectiveness of the different placements of NBSs selected in Section 3.3, the impact of the NBSs on inundation depth and extent was modeled using the 2D hydrodynamic flood model developed for the Gawler River catchment (see Section 3.2). Effectiveness was assessed using the total reduction in the level of inundation and the total reduction in the damage to building stock over the study area in response to the addition of the portfolio of NBSs. The latter was determined with the aid of vulnerability curves that determine the percentage destruction of a building based on inundation level. This percentage is multiplied by the value of the building to determine the damage. By performing this calculation using inundation levels with and without the presence of regional portfolios of NBSs, the reduction in damage in building stock was determined. The value of the building stock was determined from a report for the Gawler River [54], as were the vulnerability curves for the different building types connected to the land use classes used in this study.

The effective uniform infiltration rate that best represents the impact of the addition of different portfolios of NBSs for the selected 100 m × 100 m modeling resolution was determined using the approach outlined in Section 2.2. Different configurations of NBSs within 100 m × 200 m areas were considered for building- and block-size measures, as summarized in Figure 5. For building size measures (Scheme A, Figure 5), one spatial configuration of ten measures with a size of 15 m × 30 m and an infiltration rate of 400 mm/h across the 100 × 200 m area was considered. For block size measures (Scheme B, Figure 5), one spatial configuration of two measures with a size of 75 m × 90 m and different infiltration rates ranging from of 10 mm/h to 500 mm/h was considered. These sizes and infiltration rates were considered based on examples in Water Sensitive SA [51].

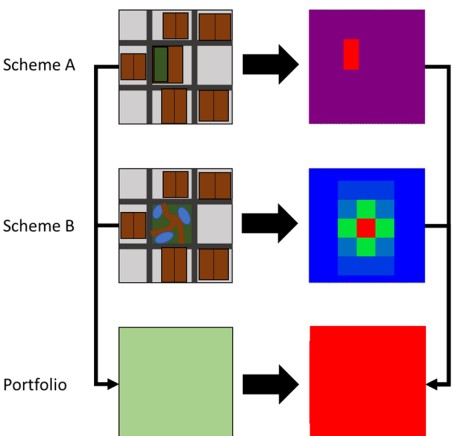

**Figure 5.** Overview of computational representation of NBS schemes.

A manual calibration process was used, iteratively adjusting the uniform infiltration rate over the 100 m × 200 m cell until the runoff hydrographs from the different configurations of building- and block-size measures closely matched those obtained when the uniform infiltration rate was used. These hydrographs were compared at nine random locations within the 100 m × 200 m cell using visual inspection.

To assess the suitability of using portfolios of NBSs at regional scales, the computational experiments summarized in Figure 6 were conducted (see Figure 4). As shown, the effectiveness of the different portfolios of NBSs considered was assessed for flood events of different return periods, including 1 in 10 (average intensity of 24.5 mm/h), 1 in 20 (average intensity of 29.7 mm/h), and 1 in 50 (average intensity of 37.3 mm/h)-year events

for a 60 min duration adjusted to fit this study from design rainfall events provided by the Bureau of Meteorology. For each of these events, the effectiveness of two different configurations of NBSs was assessed, corresponding to NBS placement locations determined using the different inundation thresholds considered (see Table 1 and Figure 4). As mentioned previously, effectiveness was assessed both in terms of reduction in inundation depth and reduction in damage to building stock. Consequently, the output of this step is a set of trade-off curves relating different degrees of coverage of the relevant region with NBSs and corresponding reduction in inundation level and building damage (Figure 4, Step 4, Outcomes).

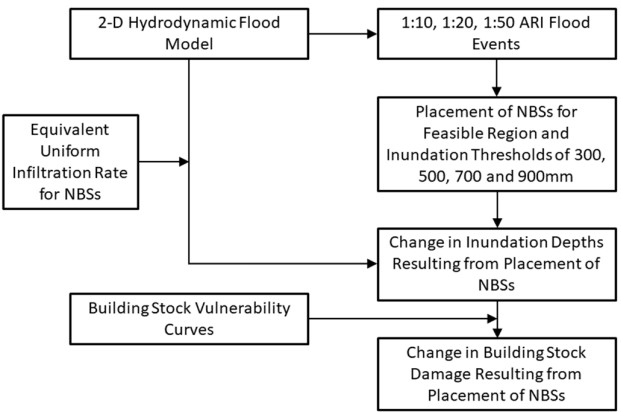

**Figure 6.** Overview of computational experiments.

## 4. Results and Discussion

In this section, the results of the computational experiments outlined in Section 3.4 are presented and discussed. First, the results of the calibration process used to determine the equivalent uniform infiltration rate that enables portfolios of NBSs to be modeled at coarser spatial scales for the purposes of regional planning (Objective 1(ii)) are presented and discussed (Section 4.1). Next, the results of the computational experiments designed to assess the degree to which portfolios of nature-based solutions can mitigate pluvial flooding at the catchment scale for the Gawler River case study (Objective 2) are presented and discussed (Section 4.2).

### 4.1. Determination of Equivalent Infiltration Rate

As part of the calibration process, a uniform infiltration rate of 100 mm/h was found to give good results, as shown in Figure 7. In this figure, representative modeled runoff hydrographs obtained at four of the nine random locations within the 100 m × 200 m spatial units considered are shown for the two detailed sets of configurations of NBSs (building size measures (Scheme A, Figure 2) and block size measures (Scheme B, Figure 2)), as well as when a uniform infiltration rate of 100 mm/h was used over the entire 100 m × 200 m spatial unit. As shown, use of an equivalent uniform infiltration rate results in very similar runoff hydrographs in comparison to modeling different configurations of NBSs in more detail. Although there is a better match to the building size measures, the discrepancy in hydrograph peak and timing compared with the block size measures is minimal when an equivalent uniform infiltration rate is used. Similar results were obtained for the other five locations (see Appendix E).

The above results suggest that the proposed approach is suitable for determining an equivalent uniform infiltration rate that enables portfolios of NBSs to be modeled at coarser spatial scales for the purposes of regional planning. The results also suggest that it is possible to adopt a relatively coarse spatial modeling resolution, such as 100 m × 100 m, for the purposes of approximating the potential impact of portfolios of NBSs that are distributed throughout a region as part of preliminary assessment studies. However, once suitable locations of portfolios of NBSs have been identified, a more finely resolved

modeling approach is likely required for the preliminary and detailed engineering design of individual NBSs.

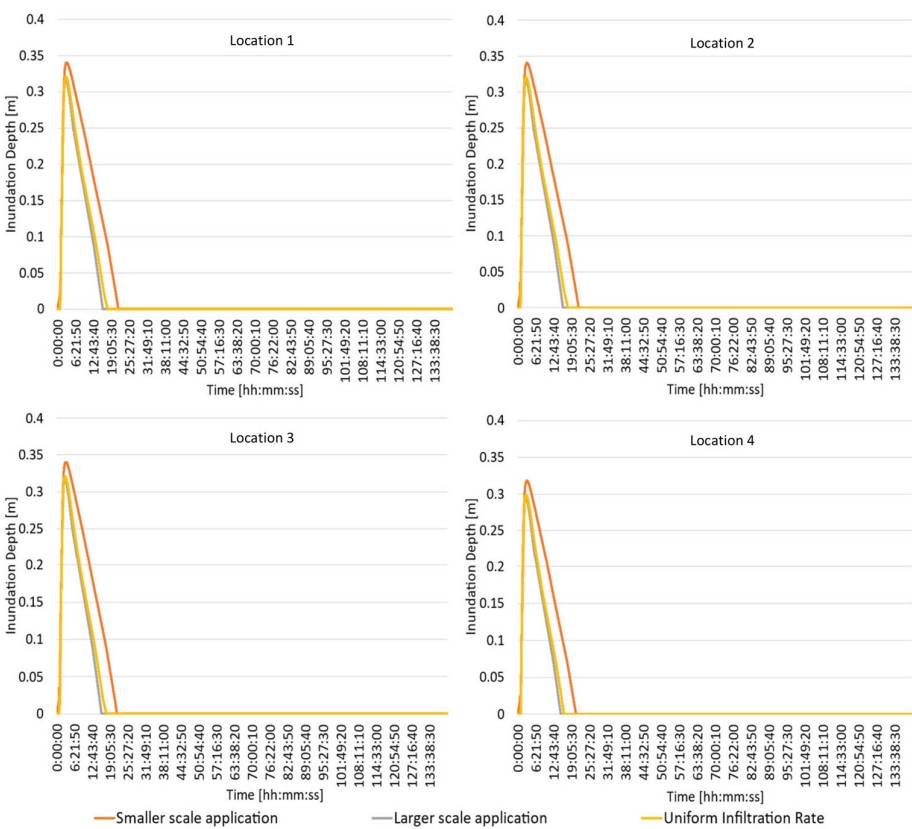

**Figure 7.** Comparison of inundation depths of different detailed nature-based solutions and the uniform infiltration rate over time in four different locations.

*4.2. Determination of Trade-Off between Number of NBSs and Effectiveness*

The trade-offs between using a larger number of strategically placed NBSs, as represented by different utilization rates of the total relevant region, and their flood risk reduction effectiveness, as represented by the reduction in damage to building stock and the reduction in flood inundation level, for different return period events (i.e., 1:10, 1:20 and 1:50) are shown in Figures 8 and 9, respectively. As can be seen from Figure 8, the strategic placement of NBSs throughout the Gawler region has the potential to reduce damage significantly. For example, for a 1:10 year event, a portfolio of NBSs covering 0.2% of the area under consideration can reduce the resulting damage by 20%. This number increases to around 32% if the area covered is 1%. However, the marginal increase in the effectiveness NBSs decreases rapidly as the area covered by NBSs increases, as shown by the highly non-linear nature of the plots in Figure 8, where the biggest marginal increase in effectiveness occurs for percentages of less than 0.15% and almost reaches zero for percentages in excess of 3.5%. This highlights the effectiveness of the approach proposed in Section 2 in terms of being able to identify locations where NBSs should be placed to achieve the biggest returns for investment.

The shape of the trade-off curves between the number of NBSs and the percentage reduction in building stock damage is very similar for events with different return periods (Figure 8). However, as expected, for the same configuration of NBSs, there is a reduction in effectiveness as the return period of the flood event increases. For example, the damage reduction associated with 0.2% coverage of the area under consideration with NBSs decreases from 20% for a 1:10 year event to around 14% for a 1:50 year event and from around 32% to around 27% when the coverage is at 1%. However, overall, the results obtained indicate that using a strategically placed portfolio of NBSs still appears to be a potentially

successful strategy for reducing the damage to building stock caused by flooding at the regional scale for higher return periods, such as 1:50 year events.

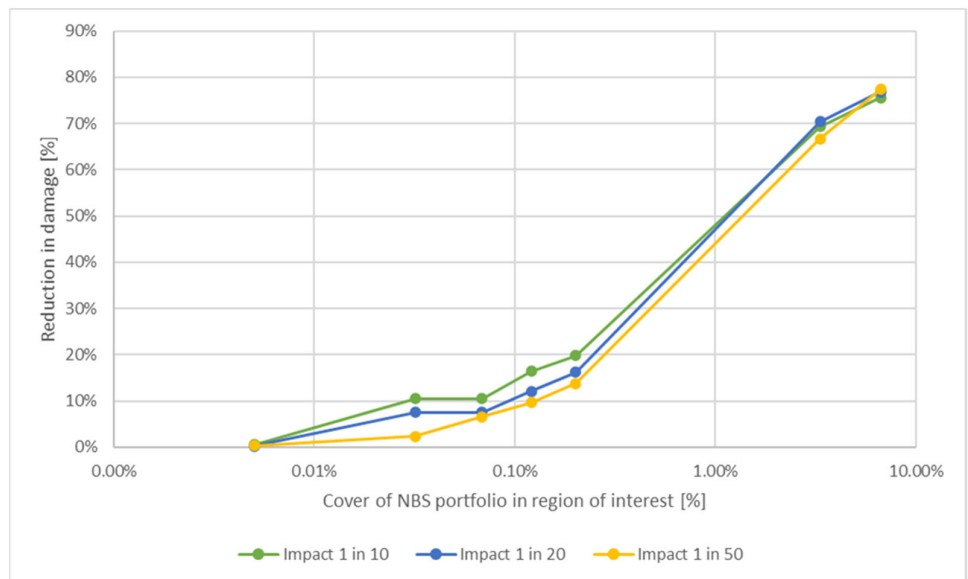

**Figure 8.** Reduction in total damage over the study area in response to the addition of NBSs for each return period and coverage in nature-based solutions.

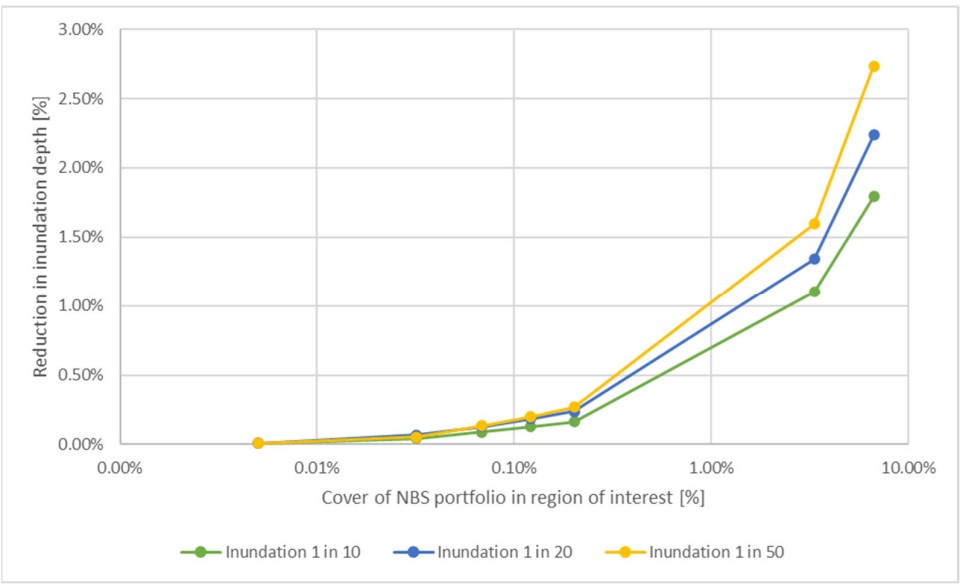

**Figure 9.** Reduction in total inundation depth over the study area in response to the addition of NBSs for each return period and coverage in nature-based solutions.

Although the placement of a portfolio of NBSs at the regional scale resulted in significant reductions in building stock damage (Figure 8), this did not result in significant reductions in inundation depths (Figure 9). A comparison of total inundation depth before and after the placement of the NBS portfolio was used to determine the rate of change. For example, for a 1:10 year event, by placing a portfolio of NBSs on 0.2% of the area under consideration, the resulting inundation depth was only reduced by about 0.16%, whereas the corresponding damage was reduced by 20%. Similarly, when the percentage coverage of NBSs was 3.34% and 6.66%, the corresponding reductions in inundation depths were only 1.1% and 1.8%, respectively, whereas the associated reductions in damage were

around 69% and 76%. These results highlight the non-linearity in the relationship between inundation depth and damage, especially the relationship between the effect of floor level height on damage (i.e., zero damage occurs if inundation depths are below floor level and potentially significant damage occurs once inundation depths exceed floor levels). However, it also confirms the ability of the approach introduced in Section 2 to identify the most promising locations for the placement of a portfolio of NBSs at the regional scale in terms of damage reduction.

As was the case for the shape of the trade-off curves between the number of NBSs and the percentage of reduction in building stock damage (Figure 8), the shape of the trade-off curves between the number of NBSs and the percentage reduction in total inundation depth is very similar for events with different return periods (Figure 9). However, contrary to a percentage change in building stock damage, the percentage changes in inundation depth increase with an increase in return period. This is because of the increase in intensity of the rainfall events. However, as discussed above, this increased percentage reduction does not translate into an increased percentage reduction in losses as the effectiveness of the NBS portfolio reduces.

Although the shapes of the trade-off curves in Figures 8 and 9 are both non-linear in the lower ranges of NBS coverage (~0% to 0.2%), this is not the case for higher ranges. While the marginal benefit of adding a larger number of NBSs decreases as the number of NBSs increases with regard to percentage reduction in building stock damage (Figure 8), as discussed above, the same is not the case for percentage reduction in inundation depths, which continues to increase with an increase in the size of the area covered with NBSs as a result of the increase in infiltration rate (Figure 9). However, if there is no building stock in these areas, there is no further reduction in building stock damage, even though there is a reduction in inundation depths. This further highlights the importance of an algorithm that enables the most beneficial locations of NBSs to be identified at the regional scale.

## 5. Conclusions

Flooding causes significant damage in urban areas worldwide. Structural options are often used to mitigate flood risk at regional scales. While the use of nature-based solutions (NBSs) is becoming increasingly popular as a potential alternative to structural measures due to their adaptability and co-benefits, such as improved water quality and increased urban amenity, their application in urban areas has only been considered at building or block scales. In this paper, we introduce and illustrate an approach that enables the utility of NBSs to be assessed at regional scales, including the ability to model the urban flood reduction benefits of NBSs at spatial resolutions that are commensurate with those commonly used in spatial planning studies (e.g., 50 m $\times$ 50 m to 500 m $\times$ 500 m) and the ability to identify the most suitable locations at which to place a portfolio of NBSs at a regional scale.

The proposed approach was applied to the Gawler River region in South Australia, which is prone to flooding that has the potential to cause significant damage in urban areas. The most relevant locations for the placement of portfolios of NBSs in order to reduce flood risk were identified and the potential benefit of using portfolios of NBSs was assessed. The results indicate that the strategic placement of a portfolio of NBSs has the potential to reduce regional flood risk significantly. For the case study considered, by placing portfolios of NBSs on 0.2% of the area under consideration, the resulting damage to building stock can be reduced by 20% for a 1:10-year event, 16% for a 1:20-year event, and 14% for a 1:50-year event. These reductions in building stock damage increase to around 32% for a 1:10-year event, 30% for a 1:20-year event, and 27% for a 1:50-year event if the area covered is 1%.

While the case study considered has demonstrated the potential of using portfolios of NBSs to reduce urban flood risk at regional scales, application of the proposed approach to a wider range of case studies is needed to better understand the conditions under which such an approach might provide potentially viable alternatives to more commonly used structural mitigation strategies. In addition, it should be noted that risk reduction is only

one of the criteria used to determine the viability of such an approach. For example, there is a need to consider a range of economic, social, and environmental criteria as part of a multi-criterion assessment to ascertain which approach to regional flood risk reduction is most appropriate in a given decision context. This includes consideration of the feasibility and acceptance of placing portfolios of NBSs over larger urban areas, which will most likely require significant stakeholder engagement. Additional allocation rules could also include the upstream placement of NBSs for a downstream impact in flood reduction. In addition, the presented approach is suitable for high-level planning investigations and more detailed modeling and design of individual NBSs is required prior to implementation. Consequently, the proposed approach and case study results presented in this paper provide the first step towards the consideration of portfolios of NBSs for urban flood risk reduction at the regional scale. However, this study does lead the way towards the consideration of an alternative approach to reducing regional flood risk in urban areas that is adaptive and has the potential to result in a range of co-benefits, such as improved water quality and amenity.

**Author Contributions:** Conceptualization, E.M.H. and H.R.M.; methodology, E.M.H., H.R.M. and A.C.Z.; validation, E.M.H. and H.v.D.; formal analysis, E.M.H.; data curation, E.M.H.; writing—original draft preparation, E.M.H.; writing—review and editing, H.R.M., A.C.Z. and H.v.D.; visualization, E.M.H.; supervision, H.R.M., A.C.Z. and H.v.D.; All authors have read and agreed to the published version of the manuscript.

**Funding:** This research was supported by a "GOstralia!/University of Adelaide" scholarship at the University of Adelaide.

**Data Availability Statement:** The data presented in this study are available on request from the corresponding author. The data are not publicly available due to the data being part of a larger PhD research study.

**Acknowledgments:** The Authors would like to thank DHI for supporting this research by providing a license for their flood model. The Authors would also like to thank Geoff Fisher and Michael Di Matteo from Water Technology for offering their flood model of the Gawler River catchment for calibration and validation purposes. Finally, the authors would like to thank the anonymous reviewers of this paper, whose comments have helped to improve the quality of this paper.

**Conflicts of Interest:** The authors declare no conflict of interest.

## Appendix A. Land Use Map Gawler River Region

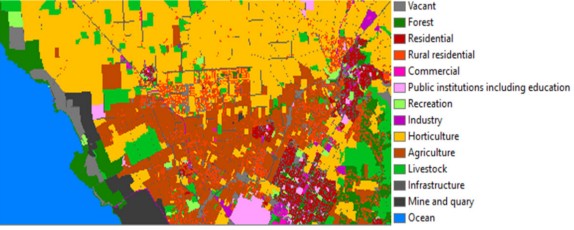

**Figure A1.** Land use map case study region.

## Appendix B. Flood Maps

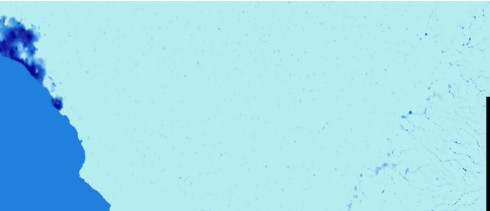

**Figure A2.** 1 in 10 rainfall event flood map.

## Appendix C. NBS Portfolio Placement throughout Gawler River Region

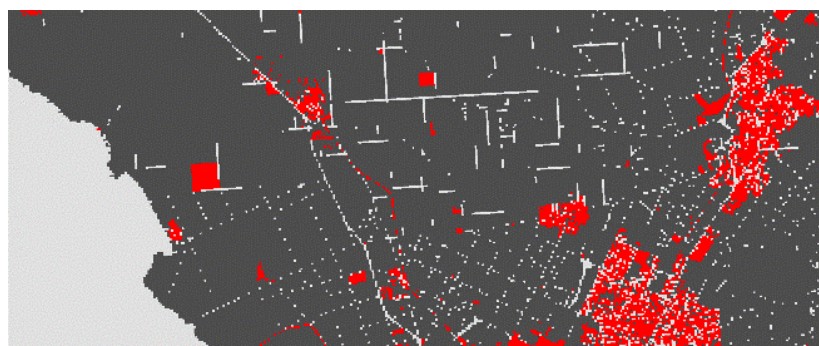

**Figure A3.** NBS potential locations.

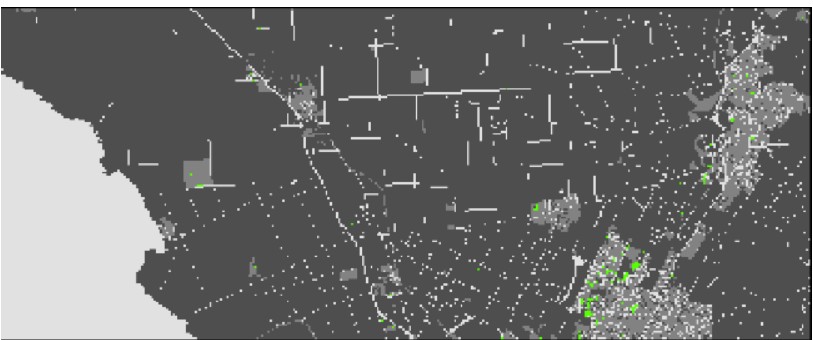

**Figure A4.** NBS relevant locations.

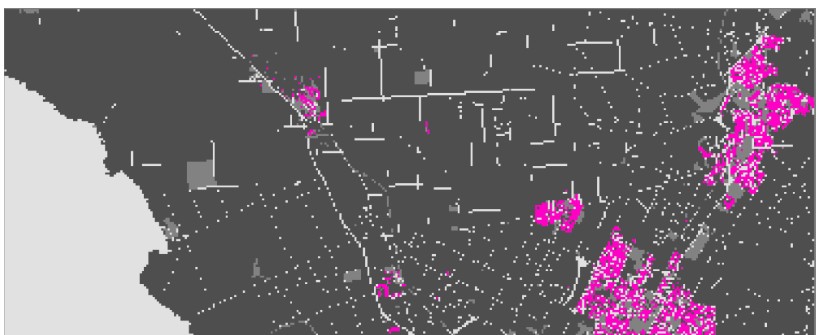

**Figure A5.** NBS portfolio placement based on all residential cells.

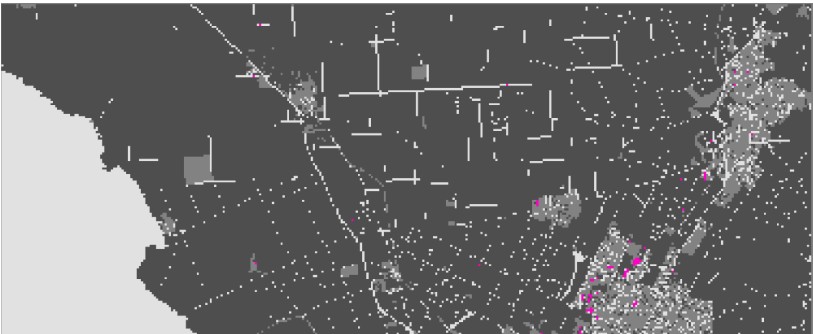

**Figure A6.** NBS portfolio placement based on 30 cm threshold.

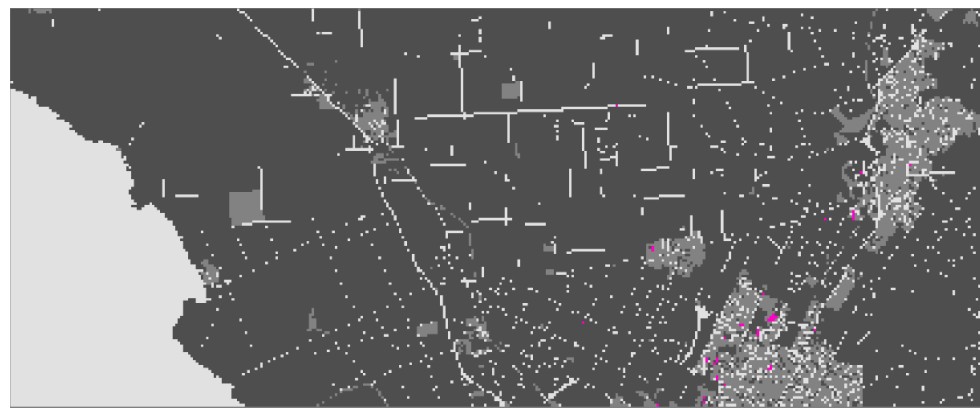

**Figure A7.** NBS portfolio placement based on 50 cm threshold.

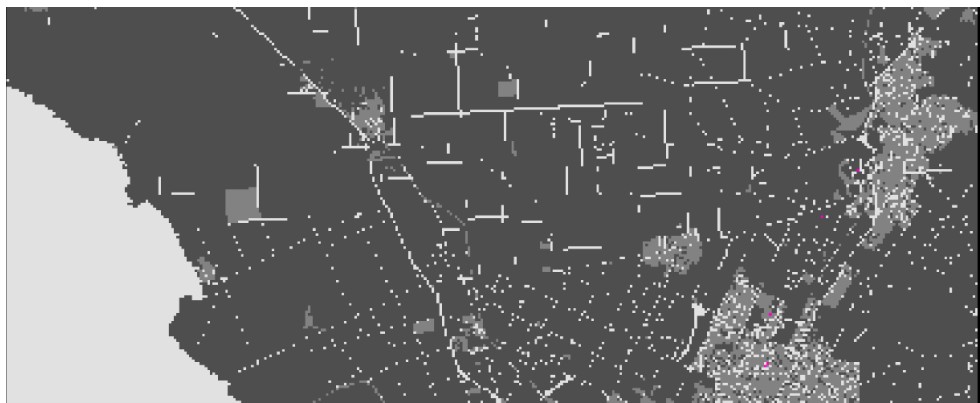

**Figure A8.** NBS portfolio placement based on 70 cm threshold.

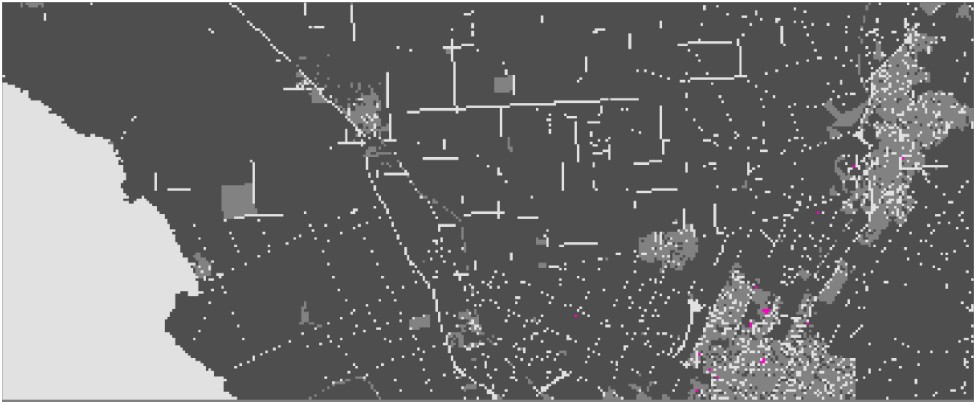

**Figure A9.** NBS portfolio placement based on 90 cm threshold.

## Appendix D. Performance of Uniform Infiltration Rate against the Two Different Schemes across the 100 m × 200 m Area

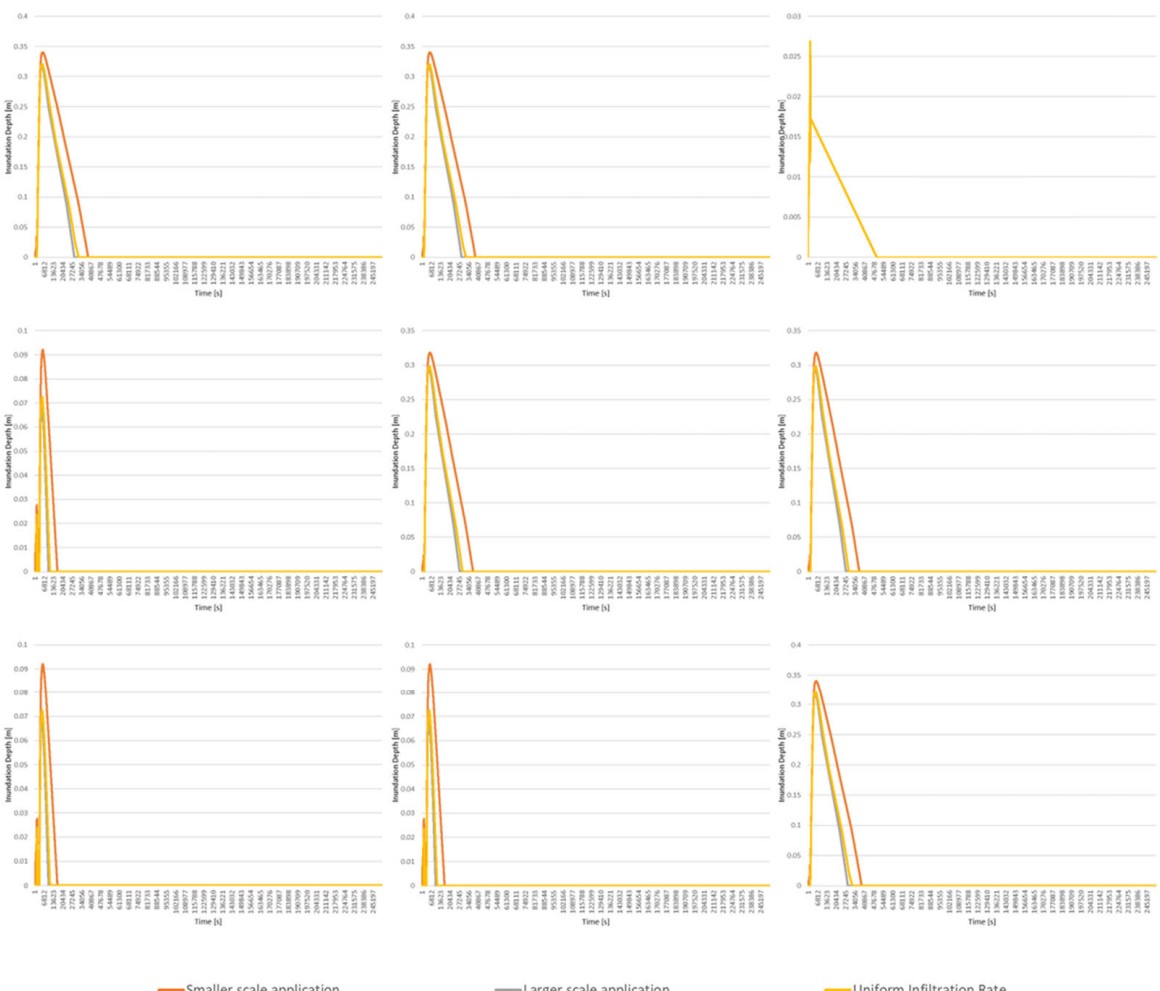

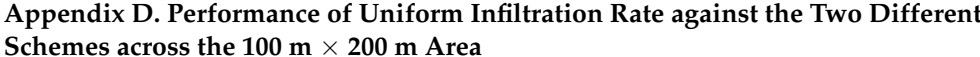

**Figure A10.** Location graphs to determine uniform infiltration rate.

## Appendix E. Flood Model Validation

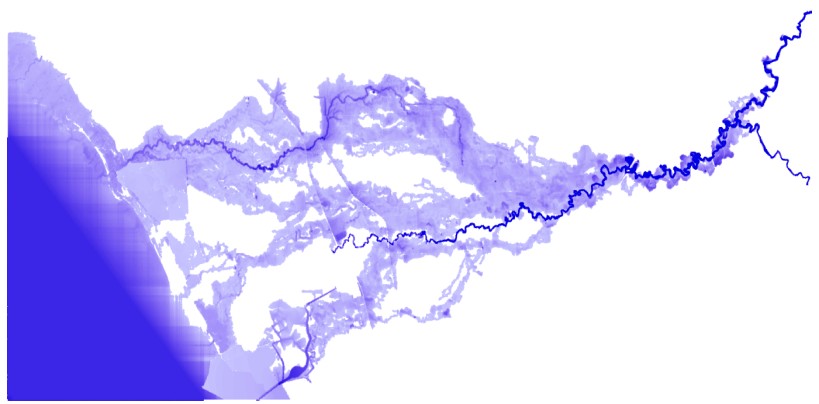

**Figure A11.** Flood map result simulated by model provided by WaterTech.

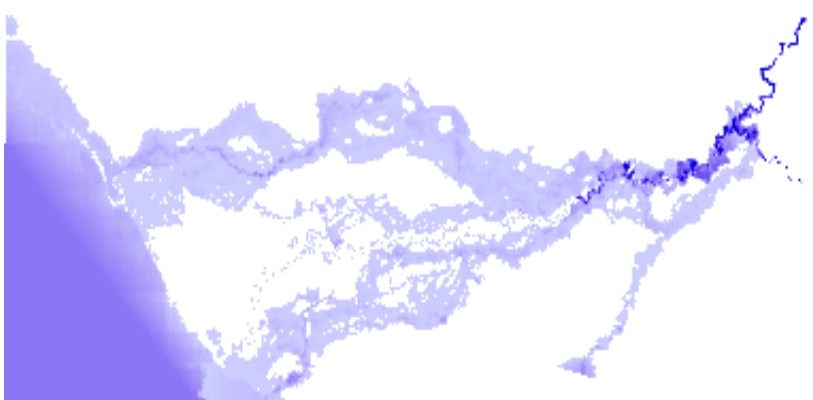

**Figure A12.** Flood map result simulated by new Mike Flood model.

## Appendix F. Impact Assessment

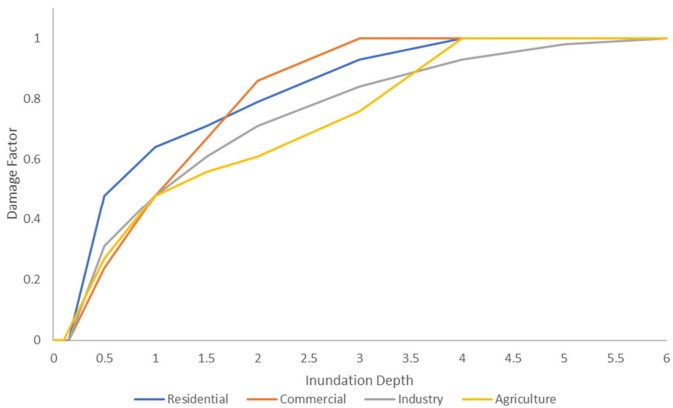

**Figure A13.** Vulnerability curves used in impact assessment.

**Table A1.** Damage factor based on inundation depth for different land use classes.

| | Damage Factor | | | |
|---|---|---|---|---|
| **Inundation Depth** | **Residential** | **Commercial** | **Industry** | **Agriculture** |
| 0 | 0 | 0 | 0 | 0 |
| 0.15 | 0 | 0 | 0 | 0 |
| 0.5 | 0.48 | 0.24 | 0.31 | 0.27 |
| 1 | 0.64 | 0.48 | 0.48 | 0.48 |
| 1.5 | 0.71 | 0.67 | 0.61 | 0.56 |
| 2 | 0.79 | 0.86 | 0.71 | 0.61 |
| 3 | 0.93 | 1 | 0.84 | 0.76 |
| 4 | 1 | 1 | 0.93 | 1 |
| 5 | 1 | 1 | 0.98 | 1 |
| 6 | 1 | 1 | 1 | 1 |

**Table A2.** Value per land use class considered in impact assessment.

| | Residential | Rural Residential | Commercial | Industry | Agriculture | Horticulture |
|---|---|---|---|---|---|---|
| Value | AUD 5,769,663 | AUD 338,349 | AUD 7,632,993 | AUD 732,208 | AUD 3053 | AUD 24,845 |

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
