# Peer review of "Effectiveness of Nature-Based Solutions for Mitigating the Impact of Pluvial Flooding in Urban Areas at the Regional Scale"

_water, doi:10.3390/w15040642_

Round 1
Reviewer 1 Report
The authors firstly modeled and evaluated the effectiveness of nature-based Solutions for mitigating the impact of pluvial flooding in urban areas at the regional scale. Generally, the causal relationship is well-established and easy to follow. It is very meaningful work in both scientific and applicable aspects since it can be useful for policymakers in developing management plans. While I have some conservation about their methods, the authors already restricted their conclusion and supporting data, especially the last part about method limitation, which provides valuable information and sticks on what their results tell and warn potential users with their discretion. Moreover, there are some structural issues. Therefore, there are still several issues that need to be addressed before considering this manuscript for publication.
Comments:
- The structure of this manuscript looks odd to me. Authors seem to propose one model and use one example to test, which makes it like a report instead of peer-reviewed manuscript. Are there any better ways to do that?
- The format is also odd. It looks like this manuscript was submitted in review mode. This kind of trivial error should be avoided and authors need to be careful and show some respect for reviewers.
- Line 156: Reference 44 should be rectified.
- Line 195: Reference 46 should be rectified.
- Line 226: Reference 48 should be rectified. Please check all over the manuscript to rectify them all.
- Please justify the reason for selecting 100 mm as the infiltration rate threshold.
- Where areFigures 7 and 8?
Reviewer 2 Report
The topic is very interesting and the methodology is correct. Similarly, the results have been presented clearly and appropriately and the conclusions are reliable and significant. However, it is based on a conceptual misconception: “NBSs are generally applied at house- or block-scales, their potential for reducing the impacts of urban flooding at the regional scale are unknown”. This does not invalidate the paper, but this statement should be reconsidered, since there are many papers on NBSs at regional scales, with very good results.
Reviewer 3 Report
The paper presents a procedure for locating Nature-Based Solutions (NBS) and evaluating their effectiveness at the regional scale in order to mitigate the impacts induced by pluvial flooding. Within the procedure, the numerical modelling step requires the evaluation of a uniform infiltration rate representing the equivalent effect of portfolios of NBS. The methodology is applied to a case study in South Australia. The topic addressed in the paper is potentially of interest, but the authors need to clarify some main issues and significantly revise the paper in order to let the reader apply the proposed procedure to different study areas. My overall recommendation is major revision.
In what follows, a list of the major points that have to be fixed is reported:
1) Line 54: the authors should briefly describe the main topics of the nine works here mentioned;
2) Line 148: the authors have to better explain the rules of allocations that are simply marked as “case study specific” and “predetermined”: my suggestion is to provide here a general strategy (or a couple of examples) and fully describe the rules adopted for the Australia case in Sect.3;
3) Line 159: please define the “performance indicators of interest” to be adopted;
4) The authors mention the concept of portfolios of NBS several times in the paper: I would suggest them to provide some examples about the content of these portfolios (e.g. green roof + parkland+ what else?);
5) Line 202: focusing on the spatial resolution adopted in the numerical simulations, it would be interesting to assess the influence of the adopted mesh size on the results (e.g. by performing a sensitivity analysis);
6) Lines 208-219: the authors have to provide more details concerning the calibration of a uniform infiltration rate. Which model was adopted to estimate the runoff hydrographs from the NBS in schemes A and B? Which model was adopted to estimate the runoff hydrographs using a unique infiltration rate? Since according to step 2 different relevant locations of NBS can be defined for a given study area, the calibration of the infiltration rate is carried out for each location?
7) The figures and the graphs appearing in the Appendixes are never described in the text and so they appear useless. My suggestion is to include these figures directly inside the sections of the paper and comment them in the text;
8) Line 277: the authors mention the adoption of GPUs to reduce the computational times. I think it would be interesting to known the computational costs of the proposed procedure (e.g. the runtimes for the calibration of the infiltration rate, the runtimes for the definition of the flooding scenarios, etc.);
9) Line 291: please show the results of the calibration;
10) Line 317, Table 1: please clarify if the “Relevant” values are referred to a 100 mm water depth threshold. Moreover the authors should better explain the reason why they selected different thresholds from 30 to 90 cm even if they did not use them for defining the relevant locations;
11) Lines 335-340: I would suggest adding a sketch representing the assumed configuration, also including the locations mentioned at line 344;
12) Figures 5 and 6: in the previous sections the methodology dealt with flooding maps and maps with relevant locations where placing NBS. However, the effectiveness of NBS is assessed in Figures 5 and 6 by means of linear graphs representing the change in damage and inundation. The authors have to describe the indicator they adopted to evaluate these changes. For example, focusing on the inundation depth, did the authors compute the difference between the water depth maps obtained with and without the equivalent infiltration rate? How did they move from a 2D map to the linear indicator in Figure 6? Additionally, it would be interesting to see the variability of the effectiveness of NBS within the study area.
In what follows, a list of the minor points that have to be fixed is reported:
1) The authors should mention in the abstract the application to a case study;
2) Line 170: I do not think that the spatial extent of the area to be modelled is one of the criteria to define the resolution. For example, when dealing with the modelling of urban flooding, the mesh size has to be defined in order to enable the simulation of local scale phenomena despite the total extent of the study area;
3) Line 251: from here on out please check the cited figures (e.g. figure 5 instead of figure 4 at line 154, etc.);
4) Line 261: relevant locations are identified by overlapping the maps of the potential locations and of the simulated water depths: please add this information also in the general description of the procedure (Sect. 2.1).
Round 2
Reviewer 3 Report
The paper presents several modifications and the authors have address most of my issues.